# A Critical Appraisal of Contemporary and Novel Biomarkers in Pheochromocytomas and Adrenocortical Tumors

**DOI:** 10.3390/biology10070580

**Published:** 2021-06-25

**Authors:** Marina Tsoli, Kosmas Daskalakis, Eva Kassi, Gregory Kaltsas, Apostolos V. Tsolakis

**Affiliations:** 11st Department of Propaedeutic and Internal Medicine, Laiko University Hospital, National and Kapodistrian University of Athens, 11527 Athens, Greece; ekassi@med.uoa.gr (E.K.); gkaltsas@endo.gr (G.K.); 2Department of Surgery, Faculty of Medicine and Health, Örebro University, 70185 Örebro, Sweden; kosmas.daskalakis@oru.se; 3Department of Surgery and Centre for Clinical Research Uppsala University, Västmanland Hospital, 72189 Västerås, Sweden; apostolos.tsolakis@regionvastmanland.se or; 4Department of Medicine Huddinge, Karolinska Institute, 14186 Stockholm, Sweden

**Keywords:** metanephrines, chromogranin A, Ki-67, molecular, genetic, micro-RNA

## Abstract

**Simple Summary:**

Pheochromocytomas/paragangliomas (PPGLs) and adrenocortical tumors are neoplasms that originate from different regions of the adrenal gland and display significant heterogeneity with respect to their biological and clinical behavior. They may be sporadic or develop in the context of hereditary syndromes. Adrenocortical tumors are mostly benign but less than 2% are carcinomas associated with a poor outcome when diagnosed in advanced disease. The majority of PPGLS are benign, but a subset may develop metastatic disease. In particular, for PPGLs, it is mandatory to identify biomarkers of high sensitivity and specificity that lead to accurate diagnosis and predict patients with a malignant potential that would benefit from aggressive surveillance and administration of early treatment. Current biomarkers include mostly histopathological and genetic parameters but none of them can be considered independent predictive factors. Recent genomic studies have implemented new molecular biomarkers of high accuracy for the diagnosis and management of PPGLs and adrenocortical tumors. In this review, we summarize the current and novel biomarkers that may be considered valuable tools for diagnosis and prediction of malignancy in patients with PPGLs and adrenocortical tumors.

**Abstract:**

Pheochromocytomas/Paragangliomas (PPGLs) and adrenocortical tumors are rare neoplasms with significant heterogeneity in their biologic and clinical behavior. Current diagnostic and predictive biomarkers include hormone secretion, as well as histopathological and genetic features. PPGL diagnosis is based on biochemical measurement of catecholamines/metanephrines, while histopathological scoring systems have been proposed to predict the risk of malignancy. Adrenocortical tumors are mostly benign, but some can be malignant. Currently, the stage of disease at diagnosis and tumor grade, appear to be the most powerful prognostic factors. However, recent genomic and proteomic studies have identified new genetic and circulating biomarkers, including genes, immunohistochemical markers and micro-RNAs that display high specificity and sensitivity as diagnostic or prognostic tools. In addition, new molecular classifications have been proposed that divide adrenal tumors in distinct subgroups with different clinical outcomes.

## 1. Introduction

Pheochromocytomas and paragangliomas (PPGLs) are rare tumors with an estimated reported annual incidence of 3–8 cases per million population per year while they account for approximately 5% of adrenal incidentalomas [1,2]. A pheochromocytoma is a tumor arising from chromaffin cells in the adrenal medulla while a paraganglioma derives from the extra-adrenal chromaffin cells of the sympathetic or parasympathetic ganglia [3]. PPGLs commonly produce one or more catecholamines, epinephrine, norepinephrine and dopamine but rarely may be biochemically silent. Parasympathetic paragangliomas do not produce catecholamines [3].

PPGLs are usually benign but may have life-threatening or devastating consequences if not diagnosed or left untreated [4,5]. Patients with sympathetic PGLs display 10 times higher incidence of cardiovascular events before their diagnosis [6]. The prevalence of malignancy, defined as the presence of metastases in non-chromaffin tissue such as lymph nodes, liver and bone, varies between 10% and 17% while the presence of germline mutations in the gene encoding succinate dehydrogenase subunit-B (SDHB) is associated with metastatic disease in 40% of cases [7,8,9]. In patients operated for a PPGL, the 5-year incidence of recurrence is approximately 4.7% while more than 40% of cases represent malignant recurrence [10]. Μetastases may be observed even 15 years after the diagnosis of a tumor initially thought to be benign [11]. Malignancy is associated with a 5-year mortality rate of 40 to 95% [12,13,14].

The risk stratification of PPGLs regarding their metastatic potential, based on clinical and histopathological findings alone, is considered particularly challenging. Several histopathological scoring systems have been proposed, including PASS (Pheochromocytoma of the Adrenal Gland Scale Score) and GAPP (Grading of Adrenal Pheochromocytoma and Paraganglioma) score but none of them can predict accurately the aggressiveness of PPGLs and the risk of metastases [15,16]. In addition, a Ki-67 labeling index (LI) above 3 to 5% has been reported to be associated with an increased risk of metastasis in PPGL [17]. However, the marked intra-tumoral heterogeneity, as well as the inter-observer variability may result to inaccurate Ki-67 LI calculation and render it an insecure predictive marker [16,18]. Hence, in the recent WHO classification, all PPGLs were proposed to have a malignant potential [19].

The tumors of the adrenal cortex are increasingly being incidentally recognized following the wide application of abdominal imaging modalities, particularly with increasing age. Most of these adrenal incidentalomas (AI) are benign adenomas [20]. However, adrenocortical tumors can cause excess steroid secretion and in less than 2% of cases, they can be malignant [21,22]. Adrenocortical carcinoma (ACC) is a rare endocrine tumor with an annual incidence of 0.7–2 cases per million inhabitants [23,24]. Symptoms may result from steroid oversecretion, compression to nearby structures and/or tumor growth and metastases [25]. The prognosis is generally poor with 5-year overall survival (OS) below 40% in most series [24,26,27]. However, the prognosis of ACC is relatively heterogeneous and significant variability in clinical presentation and outcome is observed [28].

The major prognostic factor is considered to be the tumor stage at initial diagnosis as defined by the European Network for the Study of Adrenal Tumors (ENSAT) but the risk stratification between subgroups is still variable [24,27,28,29]. In particular, the rate of recurrence after complete surgical removal in patients with disease stage I–III cannot be accurately predicted [29]. The most widely used approach for the histological evaluation of adrenocortical tumors is the Weiss score that is based on the addition of nine different histopathological features associated with tumor structure, tumor cell properties and tumor invasion [30]. However, the reproducibility of this score is limited while it has poor prognostic value in cases with borderline features (score 2 or 3) or special histological variants such as oncocytic tumors [31,32]. A recent study identified the Ki-67 LI as a prognostic factor to predict recurrence after complete surgical resection but the important intra-observer variability limits its clinical use [33].

In recent years, genomic studies have investigated gene and micro-RNA (mi-RNA) expression, chromosomal alterations and DNA methylation and have identified different subgroups of PPGLs and adrenocortical tumors with distinct molecular background and clinically relevant differences in outcome [3,12,34,35,36].

In this review, we summarize the biomarkers used to diagnose adrenocortical tumors and PPGLs, as well as new genetic and circulating biomarkers that display significant sensitivity and specificity for disease diagnosis and are associated with prognosis and risk of malignancy.

## 2. Pheochromocytomas/Paragangliomas

### 2.1. Genetics of PPGLs

A distinct feature of PPGLs is their genetic diversity. The proportion of patients with hereditary PPGLs has been estimated as high as 40% while 40–50% of these tumors have somatic mutations in one of the currently identified susceptibility genes [12,37,38,39]. Fishbein et al. showed that 27% of PPGLs have germline mutations, 39% somatic mutations (with 5–10% overlap with germline mutations), 7% gene fusions, and 89% copy number alterations [40]. The presence of these germline mutations identifies patients who are at risk for syndromic presentation, multifocal PPGL, recurrent disease and/or malignancy [41].

Whole exome sequencing is considered the new standard screening tool for genetic testing in patients with suspected hereditary PPGL but it is limited in specialized centers [36]. Hence, the clinical presentation combined with radiological and biochemical findings may serve as a guide to the appropriate genetic testing. PPGLs display considerable heterogeneity and are classified into three different groups according to patient outcome and underlying genetics [36,42]. Each subgroup is associated with distinct biochemical and clinical phenotype (Table 1) [12].

The most prevalent subtype is the ‘pseudohypoxia’ group which is characterized by somatic or germline mutations in genes involved in the tricarboxylic acid cycle, including succinate dehydrogenase subunits SDHx, fumarate hydratase (FH), von Hippel-Lindau (VHL), endothelial PAS domain 1 (EPAS1), also known as hypoxia-inducible factor 2 (HIF2A)), and in polyhydroxylases PHD1 and PHD2 [36,42]. PPGLs in this cluster are mainly extra-adrenal and aggressive, and particularly those related to SDHB mutations, are associated with the highest proportion of metastatic disease [72,73]. In addition, multiple and recurrent tumors are very common in this group that displays the poorest prognosis compared to other susceptibility gene mutations [73]. Fluorodeoxyglucose positron emission computerized tomography (FDG-PET/CT) has been proposed as a useful diagnostic tool for this group of PPGLs, probably due to genotype-related changes in energy metabolism [74].

The second subtype is the ‘kinase signaling’ group that is associated with genetic or somatic mutations in the RET proto-oncogene (syndrome MEN 2A, 2B), neurofibromin 1 (NF1), transmembrane protein 127 (TMEM127), MYC-associated factor X (MAX), kinesin-like protein (KIF1BB), receptor tyrosine kinase (MET), GTPase, and Harvey rat sarcoma viral oncogene homolog (HRAS) [36,42]. PPGLs in this group are predominately located in the adrenals and rarely develop metastatic disease [72].

The ‘Wnt-signal’ type involves somatic mutations in cold shock domain containing E1 (CSDE1), thalassemia/mental retardation syndrome X-linked (ATRX) and mastermind-like transcriptional coactivator (MAML3). It is the most prevalent genotype between sporadic PPGLs and it is frequently associated with distant metastasis or recurrence, particularly in cases of MAML3 gene fusions [40].

A systematic review demonstrated that 17% of SDHB and 8% of SDHD carriers developed metastases [75]. Welander et al., reported the frequency of metastases in hereditary PPGLs: RET, 2.9%; VHL, 3.4%; SDHD, 3.5%; and SDHB, 30.7% [76]. In a recent meta-analysis, it was shown that although molecular categorization according to SDHB provides independent information on the risk of metastasis, driver mutations status does not seem to correlate with OS [72].

### 2.2. Biochemical Diagnosis of PPGLs

The majority of clinical features of PPGLs are secondary to the secretion of catecholamines (epinephrine, norepinephrine, and dopamine) while the type and pattern of secretion are variable and determines the clinical presentation [60,77]. All patients with symptoms and signs suggestive of a PPGL should be submitted to biochemical testing for catecholamine excess independently from the presence of hypertension or other symptoms attributed to catecholamine excess [78]. Biochemical testing is also warranted in patients with adrenal incidentalomas, as well as in patients with genetic mutations in one of the susceptibility genes known to be associated with PPGLs [79,80]. In addition, screening for a PPGL should be considered in young patients with arterial hypertension, in those requiring three or more drugs for hypertension control and in cases of unexplained variability of blood pressure or of symptoms provoked by surgery, anesthesia, or drugs that may precipitate a crisis (Table 2) [78].

The classic biochemical tests have included the measurement of plasma and urinary free metanephrines (metanephrine and normetanephrine-MN/NMN) and/or urinary and plasma catecholamines [39]. The Endocrine Society guidelines recommend that values three to four times higher than the upper reference limit are almost always diagnostic for PPGLs [78]. The most frequent methods of analysis are liquid chromatography with electrochemical detection (LC-ECD), liquid chromatography with tandem mass spectrometry (LC–MS/MS) or immunoassay methods [81,82]. The LC-MS is considered the gold standard method of analysis due to its accuracy and reproducibility but it is not widely available [83]. Particularly, plasma free metanephrines are associated with a sensitivity of 97% and a specificity between 80 and 100% for the diagnosis of pheochromocytoma [84,85]. Comparably high sensitivity (97%) and specificity (91%) has been reported for the 24 h urine fractionated metanephrines [86]. A recent meta-analysis compared the accuracy of plasma and urine metanephrines, as well as the various methods of analysis and sampling [87]. Supine plasma sampling has been proved more sensitive in tumor detection and displayed higher specificity compared to 24 h urine samples [87].

The measurement of the dopamine metabolite 3-methoxytyramine (3-MT) has been introduced for the identification of patients with malignant PPGLs, head and neck PPGLs and tumors with mutations in SDHx subunits [88,89,90]. A recent study that involved 1963 patients with suspicion or at risk for PPGL, showed that the additional measurement of 3-MT only modestly increased the diagnostic accuracy of the standard measurement of MN and NMN [91]. However, the measurement of 3-MT is valuable for the discrimination of true positive and false-positive results, as it was observed that increased plasma concentrations of two or more metabolites was strongly predictive of a PPGL.

Several medications interfere with measurement techniques (acetaminophen, mesalamine, sulfasalazine in LC-ECD method) or with the disposition of catecholamines (caffeine, tricyclic antidepressants) and may result to falsely elevated levels of plasma or urinary catecholamines and their metabolites [78]. In addition, some foods, such as banana and nuts that contain high concentrations of biogenic amines, can produce false positive test results, especially when measurements include 3-MT [92]. Physical stress (e.g., critical illness) may also be associated with an increase in plasma or urine metanephrines. Therefore, ideally, the biochemical testing for a PPGL should be performed after discontinuation of all medications and substances that could result in alterations of the results.

Besides the diagnosis and follow-up of PPGL, the biochemical phenotype of a tumor may be associated with specific causative genes and syndromes. Particularly, the biochemical profile of PPGLs can be classified into the following phenotypes [36]:Truly biochemically silent phenotype: Truly silent PPGLs are, in the majority of cases, observed in the head and neck region and are often associated with SDHx syndromes.Biochemically pseudo-silent phenotype: PPGLs that belong to this category, produce catecholamines but their levels in plasma or urine are normal or near-normal due to small tumor size or fluctuations in catecholamine production.Noradrenergic phenotype: This category is characterized by increased levels of norepineprhine (NE)/NMN and is commonly associated with extra-adrenal PPGLs and the pseudohypoxia type that involves mutations in VHL and SDHx genes [36,93,94]. They are related more frequently with sustained hypertension and tachycardia rather than with paroxysmal symptoms [95].Adrenergic phenotype: PPGLs of this category produce epinephrine (E)/MN and/or NE/NMN [93,94]. They are commonly located in the adrenals are epinephrine associated with the kinase signaling group that involves RET and NF1 mutations [36,40]. These tumors are related more frequently with paroxysmal symptoms, often developed after concomitant use of medications, anesthetics, and tyramine-rich food while patients with this phenotype may also develop hyperglycemia and hyperlipidemia [36].Dopaminergic phenotype: This category includes PPGLs that produce dopamine/3-MT while the levels of catecholamines and metanephrines are normal or near-normal [90,96]. They are located mainly in the head and neck area and are observed in 65% of cases with SDHx mutations, particularly in SDHB [88,90]. In addition, dopamine secretion is considered a potential predictor of metastatic disease [72].

The chromogranin A (CgA) has been associated with almost all types of neuroendocrine neoplasms including the PPGLs [97]. CgA is co-stored and co-released with catecholamines and is physiologically secreted via exocytosis by both functioning and non-functioning tumors while a positive relationship between CgA levels and tumor mass has been reported [98,99,100,101]. It has been shown to be of value in combination with urinary MN, as well as a second line test in patients with equivocal MN [102,103]. A study evaluating the clinical utility of CgA in comparison with NMN in patients with SDHB or SDHD-related PPGLs, suggested that CgA is at least as a sensitive biomarker of the disease as the NMN and can be routinely added to the biochemical work-up [104]. It is of note that this study also showed a lack of efficacy of CgA in patients with parasympathetic head and neck paragangliomas. Furthermore, the circulating concentration of CgA fragments, such as WE-1, EL35 and GE-25, has been found to be increased in patients with PPGLs, but its biological activity and utility as tumor markers have not been addressed [105].

Falsely elevated CgA concentrations may be observed in patients treated with proton pump inhibitors (PPIs) or other acid-blocking medications, but this effect is fully eliminated after discontinuation of the medication for two weeks [97,106]. Chronic renal insufficiency, chronic gastritis, myocardial infarction and multiple other pathologic conditions may also be associated with increased CgA levels and positive results should be interpreted with caution in such cases [107,108,109].

### 2.3. Molecular Biomarkers and Future Directions

Besides the genetic analysis and the past medical history of hereditary forms of PPGLs, none of the above-mentioned biomarkers can be used as independent predictive marker nor can be used as a therapeutic target. A recent meta-analysis suggested that a comprehensive molecular approach is probably required to cover up the inability of current histological algorithms to accurately stratify the PPGLs according to the risk of metastases [110]. Granberg et al., suggest that a combination of histological examination and molecular predictive markers may aid the physicians to pinpoint the metastatic potential of a PPGL in order to ensure individualized management and follow-up [111]. Therefore, future trials and investigations on new molecular biomarkers and novel therapeutic targets are required.

mi-RNAs are small non-coding RNAs that regulate gene expression at the post-transcriptional level. Multiple recent studies have shown a link between miRNAs and cancer development and investigate the miRNA profiling as a potential diagnostic, prognostic, and therapeutic tool. miR-210 has been found to be over-expressed in pseudohypoxia related PPGLs harboring SDHB and VHL mutations [112,113]. In addition, a recent study demonstrated that lower serum expression levels of miR-210 were associated with malignant PPGLs [114]. miR-483-5p and miR-101 were found to be up-regulated in malignant PPGLs while the level of miR-101 was higher in SDHD-related tumors [115,116]. In addition, the tumor suppressor miR-15a and miR-16 were under-expressed in malignant PPGLs vs. benign tumors [117]. A recent study identified a signature of six miRNAs that was associated with metastatic risk and time to progression while miR-21-3p levels correlated with mTOR pathway activation and could serve as a predictive marker for mTOR inhibitor sensitivity in PPGLs [118].

Furthermore, multiple recent studies have identified long intergenic noncoding RNAs (lincRNAs) and exosomal double-stranded DNA as potential markers for diagnosis or prognostication in PPGLs [119,120,121]. However, further investigation is required until the widespread clinical use of these promising tissue and circulating biomarkers.

## 3. Adrenocortical Tumors

### 3.1. Cortisol Secretion

Increased steroid secretion may be observed in 50–75% of cases of ACC, mostly involving corticosteroids but also mineralocorticoids, androgens and estrogens [122,123,124]. Berruti et al., suggested that in 72 patients with ACC, cortisol secretion was an adverse prognostic factor [124]. Abiven et al., showed that cortisol hypersecretion was an independent prognostic factor associated with shorter overall survival (OS) and that post-operative treatment with 1,1-dichlorodiphenildichloroethane (o,p’DDD) was beneficial in patients with cortisol-secreting ACC [123]. However, other studies have failed to demonstrate a significant prognostic value of cortisol secretion in patients with localized or metastatic ACC [33,125].

The poorer prognosis of cortisol-secreting tumors may be attributed to co-morbidity of Cushing’s syndrome [126] while the immunosuppressive effect of excess cortisol production may blunt the cellular immune response and favor tumor recurrence and the development of metastases [127]. Alternatively, the biology of cortisol-secreting tumors may be associated with development of more aggressive neoplasms [123].

### 3.2. Molecular Biomarkers and Future Directions

Recent progress in genomic studies has allowed the molecular analysis of adrenocortical tumors and the identification of distinct subgroups with different outcomes. Despite the fact that the majority of studies involve a limited number of patients, several molecular biomarkers have been proved useful in the diagnostic and prognostic workout of adrenocortical tumors and is expected to become part of the routine clinical practice in the future [128].

The first study that evaluated the genomic profile of ACC was published by Giordano et al. who identified approximately 90 genes that are differently expressed in ACC compared with benign tumors [129]. Particularly, the study evaluated the expression of insulin-like growth factor 2 (IGF-2) with the presence of ACC. De Fraipont et al., demonstrated that high expression of a cluster of genes including IGF-2 and low expression of a cluster of genes associated with steroidogenesis was generally seen in ACC and was related with an increased rate of tumor recurrence [130]. Further analysis of the transcriptome of ACC identified two groups with different prognosis: the C1A group, that includes genes involved in cell cycle and consists of aggressive tumors and the C1B group that includes genes associated with cell metabolism, apoptosis and differentiation and consists of indolent tumors [131]. Of note, this transcriptome-based classification predicted OS independently of tumor stage or grade [131,132].

Recent studies have identified several miRNAs differentially expressed between ACC and normal adrenal or benign tumors [133]. The most frequently reported overexpressed miRNAs in ACC were miR-483-5p and -3p, miR-503, miR-210 and miR-184. Interestingly, the gene encoding the miR-483-5p is within the IGF-2 locus and a positive correlation has been described between this mi-RNA and IGF-2 expression [134]. It has also been observed that mi-RNA expression may be associated with prognosis. Low expression of miR-483-5p combined with enhanced expression of miR-195 may predict poor prognosis in ACC [115,133] while increased expression of miR-210 is associated with aggressive behavior and poor prognosis [35,135].

Furthermore, recent investigation on diagnostic and prognostic biomarkers of adrenocortical tumors focuses on DNA markers, including chromosomal alterations and DNA methylation profiles [136,137]. Zheng et al., have identified three subgroups of ACC based on chromosomal alterations but their prognostic value has not been validated against other prognostic factors yet [34]. Additionally, a recent study demonstrated that DNA hypermethylation of four genes (PAX5, PAX6, PYCARD, GSTP1) was strongly correlated with disease-free survival (DFS) and OS [138].

Recently, integrated pan-genomic studies have been performed, one from the ENSAT network and one from The Cancer Genome Atlas (TCGA) consortium and have implemented a new classification with distinct molecular subgroups that are associated with very different outcome [34,35]. In particular, the tumors in the C1A cluster are associated with poor prognosis and display distinct methylation, mRNA, and miRNA expression patterns, more frequent mutations in driver genes, and higher global mutation rate than tumors belonging to the C1B cluster that is associated with a better outcome [34,35].

There are also a few studies that perform proteomic analysis in an attempt to identify diagnostic and prognostic markers of adrenocortical tumors. Kjellin et al., confirmed that the IGF-2 growth factor is overexpressed in ACC compared to benign tumors [139]. A recent study showed that nuclear over-expression of pituitary tumor transforming gene (PTTG1) was only observed in ACC and never in benign tumors or normal adrenal glands while it was correlated with Ki-67 LI [140]. Poli et al., identified fascin 1, a protein involved in cytoskeletal organization, as a potential malignancy marker as it is over-expressed in ACC compared to normal adrenals [141].

Furthermore, liquid biopsy, a novel technique of personalized medicine, involves the detection of circulating tumor cells (CTCs), miRNAs, exosomes and cell-free DNA of tumor origin (ctDNA), and has been recently applied as a diagnostic and prognostic tool in ACC [136]. The presence of CTCs of adrenal origin has recently been observed in patients affected by ACC compared to benign adenomas [142]. In addition, circulating levels of mi-RNAs may be used as diagnostic or prognostic markers. It has been shown that patients with ACC have increased serum levels of miR-483-5p, miR-100, and miR210 and down-regulated miR-195, miR-335, and miR-376a compared to patients with adrenal adenomas [133]. Furthermore, it has been reported that high circulating levels of miR-483-5p and low circulating levels of miR-195 are related to shorter DFS as well as OS [143].

Altogether, these findings suggest the important role of molecular alterations in the biology and pathogenesis of adrenocortical tumors, but further multicentric studies are required to validate their diagnostic and prognostic value and to delineate their role as potential therapeutic targets.

## 4. Conclusions

PPGLs and adrenocortical tumors are complex neoplasms that require the combination of clinical, genetic, and molecular features to be properly characterized. Current standard biomarkers include measuring hormonal secretion and genetic alterations. However, these biomarkers have been investigated in small series, mostly of retrospective design, exhibit major limitations, and there is an unmet need for high accuracy biomarkers for the diagnosis and management of these tumors. In an attempt to explore new biomarkers, novel research focuses on genomic and proteomic analysis and different molecular subgroups have emerged with distinct outcomes [36,131]. As a result, large multi-centric studies are required to clarify the diagnostic and prognostic value of new tissue and blood-based biomarkers, and to delineate their potential value in clinical practice.

## Figures and Tables

**Table 1 biology-10-00580-t001:** Genes associated with PPGLs.

Gene	Subtype	Germline/Somatic	Predominant Site	Contribution to Malignancy
SDHA [43]	Pseudoxypoxia	Germline	PGL	12%
SDHB [44,45,46,47,48,49]	Pseudoxypoxia	Germline	PGL	29–73.8%
SDHC [50]	Pseudoxypoxia	Germline	PGL	Low *
SDHD [44,46,48]	Pseudoxypoxia	Germline	PGL	0–31.3%
SDHAF2 [51]	Pseudoxypoxia	Germline	PGL	0–2%
VHL [44,52,53]	Pseudoxypoxia	Germline/Somatic	PHEO	1.6–7.7%
FH [54]	Pseudoxypoxia	Germline	PHEO, PGL	Low *
EPAS1 [40,55,56]	Pseudoxypoxia	Germline/Somatic	PHEO, PGL	Low *
PHD1 [40,57]	Pseudoxypoxia	Germline	PHEO, PGL	n.a.
PHD2 [40,57]	Pseudoxypoxia	Germline	PHEO, PGL	n.a
RET [44,58,59]	Kinase Signaling	Germline/Somatic	PHEO	0–4.1%
NF1 [40,60,61,62]	Kinase Signaling	Germline/Somatic	PHEO	0–10%
MAX [43,63]	Kinase Signaling	Germline	PHEO	9–10.5%
TMEM127 [43,64]	Kinase Signaling	Germline	PHEO	5–10%
HRAS [40,65]	Kinase Signaling	Somatic	PHEO	0–6.3%
KIF1B [66,67]	Kinase Signaling	Germline	PHEO	Low *
MAML3 [40,68]	Wnt Signaling	Somatic	PHEO, PGL	Low *
CSDE1 [40]	Wnt Signaling	Somatic	PHEO, PGL	Low *
MET [69]	Not classified	Somatic	PHEO, PGL	n.a.
TP53 [40]	Not classified	Somatic	PHEO, PGL	n.a.
FGFR1 [40,70]	Not classified	Somatic	PHEO, PGL	n.a.
ARNT [40]	Not classified	Somatic	PGL	n.a.
MYCN [71]	Not classified	Somatic	PHEO, PGL	n.a.

PHEO: Pheochromocytoma, PGL: Paraganglioma, n.a.: not available.; * When the sample size of the studies is small, a quantitative estimate of the contribution to malignancy is reported.

**Table 2 biology-10-00580-t002:** Indications for biochemical testing for PPGL.

Signs and Symptoms Suggestive of PPGLs
Arterial hypertension not controlled with ≥3 anti-hypertensive drugs
Unexplained variability of blood pressure
PPGLs symptoms provoked by anesthesia, surgery, or drugs
Adrenal incidentaloma
Predisposition for hereditary PPGL or syndromic features suggesting hereditary PPGL

PPGLs: Pheochromocytomas/Paragangliomas.

## Data Availability

Not applicable.

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
