# Peer review of "A Critical Appraisal of Contemporary and Novel Biomarkers in Pheochromocytomas and Adrenocortical Tumors"

_biology, 2021, doi:10.3390/biology10070580_

Round 1
Reviewer 1 Report
Page 2, lines 567-58, “malignancy is mainly related to mortality”. This is an aiming error. Of course, malignant disease increase the risk of death.
Page 2, lines 62-63, “PPGLs reccurrence rate is approximately 1 in 100 patients-years”. This figure seems low, since up to 50% of paragangliomas are malignant.
Page 3, lines 119-120 and table 1: The 3 subgroups should be defined in Table 1.
Page 6, Section 2.3, Molecular BIomarkers and Future Directions.
This section has to be rewritten. In lines 252-254, the authors claim that “there is 252 yet no combined risk stratification model based on histology and the overall mutational 253 profile of the tumor”. However, in a recent article by Granberg et al, JCEM, 2021, Vol. 106, No. 5, e1937–e1952, one section discusses pathology, genetics and prognostic factors in PPGL. This article should be referred.
There are a few spelling errors.
Reviewer 2 Report
Review article titled "Biomarkers in Pheochromocytomas and Adrenocortical Tumors" was submitted for publication Biology journal. Author followed all instructions necessary for prepare review paper. Paper is also their milestone for further direction of investigations.
If I were author, I would prefer more attractive title, for audience, manuscript deserve it. Maybe the objective could also be like teaser for readers.
Well done.
Round 2
Reviewer 1 Report
The paper is now improved and now for publication.